# On- and Offline Multi-agent Reinforcement Learning for Disease Mitigation using Human Mobility Data

**Sofia Hurtado**
University of Texas at Austin
Austin TX, 78712
slhurtad@utexas.edu

**Radu Marculescu**
University of Texas at Austin
Austin TX, 78712
radum@utexas.edu

## Abstract

The COVID-19 pandemic generates new real-world data-driven problems such as predicting case surges, managing resource depletion, or modeling geo-spatial infection spreading. Though reinforcement learning (RL) has been previously proposed to optimize regional lock-downs, the availability of mobility tracking data with offline RL allows us to push decision making from the top-down perspective (i.e., driven by governments) to the bottom up perspective (i.e., driven by individuals). Rather than predicting the outcome of the outbreak, we utilize offline RL as a tool, along with epidemic modeling, to empower collaborative decision-making at the individual level. In our investigations, we ask whether we can train the population of a city to become more resilient against infectious diseases? To investigate, we deploy a 'city' of 10,000 agents loaded with real visits at Points of Interest (POIs) (e.g., restaurants, gyms, parks) throughout Austin, Texas during the COVID-19 pandemic (July 2020). Using a standard disease compartmental model, we find that the city of trained agents can reduce disease transmissions by 60%. This opens a new direction in using offline RL as a springboard to further the research at the intersection of artificial intelligence and disease mitigation.

Offline Reinforcement Learning Workshop at Neural Information Processing Systems, 2022.

# 1 Introduction

Though large populations, bustling commerce, and inter-regional travel mark the success of a modern society, these conditions offer a favorable environment for infectious disease spreading [1]. Considering the vulnerabilities of big cities to disease outbreaks, we ask whether we can train a population of agents to optimize their visits at POIs, (e.g., restaurants, gyms, parks, etc.) to collaboratively mitigate the disease spread.

With access to Foursquare location tracking data that logs real visits and dwell times, we load a large population of RL agents with decisions made by real people during the COVID-19 pandemic. Because we do not have access to the health status of the anonymous individuals within the Foursquare dataset, we fill in the gap by simulating a viral outbreak. Thus, we propose using a hybrid *offline* multi-agent RL and an *online* epidemic simulation model to optimize human mobility under a viral attack.

We organize the paper as follows: Section 2 discusses previous work, Section 3 outlines the approach, Section 4 shows preliminary results, and Section 5 concludes with a discussion.

# 2 Previous Work

Early in the COVID-19 pandemic, hospitals deployed a cohort model that sectioned off health care providers and patients to reduce population mixing [2]. Schools then followed suit by organizing student-teacher cohorts to reduce disease spreading [3]. If one cohort experiences an outbreak, the others can continue functioning without going into a full lockdown. In this paper, we propose pushing this cohort paradigm to highly dynamic systems (e.g., population in a city) by training RL agents to self-organize into mobility cohorts where we can incentivize 'Infectious' people to frequent different locations from the 'Susceptible' people at any given moment.

At a large scale, in response to COVID-19, governments enacted regional shut downs and travel bans that aimed to reduce population mixing. Though successful in reducing cases, the cost of maintaining long term lock-downs led to pandemic fatigue [4]. As a means to manage economic and social costs, Kompella et. al. introduce RL to optimize disease mitigation mandates at the government level [5]. In their work, the agent (i.e., government) decides to manage a city while under a disease attack.

Though helpful in advising decision making at the macro-scale, we are rather interested in informing *distributed decisions* at the micro-scale (i.e., individual level). We envision an anti-fragile society whose individuals can continue their daily lives while collaboratively avoiding infection hot-spots. By self-organizing into mobility cohorts, people can section off avenues for disease spread, thereby mitigating a disease without the need for a complete shutdown. To this end, we investigate the feasibility of using on- and offline multi-agent RL to mitigate a highly infectious disease.

# 3 Approach

We combine offline multi-agent RL with online epidemic simulation to train a population of agents to reduce the disease spread. We utilize the Foursquare dataset which consists of visits from over 36,000 individuals to various POIs throughout a metropolitan area during the 2020 pandemic (i.e., for each anonymous person, we know when, where, and how long they visit a location) [6]. For example, a 'visit' entry in the dataset consists of a device's anonymous identification, the location (POI), time of visit (hour), and how long the visit lasts (in seconds). This highly granular 'visit' entry is then aggregated to make up an agent's destination queue.

## 3.1 Epidemic Model

We apply the SEIR model [7] to individuals where an agent moves from the initial *Susceptible* state to the *Exposed* state when coming into contact with *Infectious* individuals. We then transition a *Susceptible* person to incubating when they visit a POI where the *Infectious* population density exceeds their immunity $\delta$ threshold (equation 1). After an agent is incubating, they transition to being *Infectious* after the incubation period (5 days), and to *Recovered* state after an illness period (7 days).

$$\frac{\#infections_{POI}}{total\ people} \geq \delta \tag{1}$$

## 3.2    RL Problem Definition

We define the RL problem as follows. The 'environment' consists of the POIs within a city. Each agent is loaded with destination queues pulled from the Foursquare visits dataset. At each time step (e.g., hour), the agents can choose from three actions, namely *'go to location'*, *'stay at home'*, or *'choose a safer location'*. We define the reward functions for each health status as a composition of sub-reward functions: $R_{exposure}$ (equation 2), $R_{fatigue}$ (equation 3), and $R_{global}$ (equation 4). The $R_{exposure} \in [0, 1]$ is meant to incentivize agents to reduce exposure to infections with respect to their own immunity threshold $\delta \in [0, 1]$. For example, a *Susceptible* agent with a higher immunity $\delta$ receives less of a penalty for frequenting POIs with more $\#infections_{POI}$ (number of infections at a POI) than an agent with a lower immunity threshold. The $R_{fatigue} \in [0, 1]$ is meant to incentivize agents to socially cooperate by changing their intended behavior with respect to their own fatigue parameter $\alpha \in [0, 1]$. For example, an agent with a high pandemic fatigue $\alpha$ will be penalized for choosing to deviate from their intended visits by 'staying at home' or 'going to safer location' more times than the threshold $\alpha$ allows. To keep track of the number of deviations, at each timestep, $T$, we calculate the cumulative $\#deviations$ and $\#actions$ from the beginning of the episode $t$. The $R_{global} \in [0, 1]$ is meant to motivate agents to socially cooperate if the population's *global* $infections_t$ are high at each time step $t$, even if they are not personally getting exposed.

$$R_{exposure} = \frac{1}{(1 + \#infections_{POI}) \times (1 - \delta)} \tag{2}$$

$$R_{fatigue} = (1 - \alpha) - \frac{\#deviations_{t:T}}{\#actions_{t:T}} \tag{3}$$

$$R_{global} = \frac{1}{global\ infections_t} \tag{4}$$

The rewards for each health status are defined in Table 1. We incentivize *Susceptible* agents to take into account their risk of exposure at a POI with respect to their own willingness to change behavior for the social good. When *Infectious*, they no longer worry about being exposed, but instead, they keep track of the number of people they directly infect (*#infectees*) to weigh against their respective pandemic fatigue $\alpha$. The *Incubating* and *Recovered* reward functions are similar, as these agents do not worry about being exposed.

| Health Status | Reward |
|---|---|
| Susceptible | $R_{exposure} \times R_{fatigue} + R_{global}$ |
| Incubating | $R_{fatigue} + R_{global}$ |
| Infectious | $R_{fatigue} - \#infectees + R_{global}$ |
| Recovered | $R_{fatigue} + R_{global}$ |

Table 1. Reward Functions for each Health Status

We deploy the standard REINFORCE [8] algorithm on each agent and implement policy approximation with a two layer sequential neural network. In each episode, we seed a *Susceptible* population with 10% *Infectious* agents. We terminate the episode when the virus has no one left to infect. Though our RL experiment is online, where agents can infect each other, they are incapable of visiting locations outside of their destination queues (hence the offline aspect of our approach).

## 4    Preliminary Results

As a proof of concept, we evaluate our approach by simulating an infection throughout a naive vs. trained population of 10,000 agents. The naive population exclusively follows the Foursquare destination queues rather than a learned policy. In Figure 1a, the simulated disease cumulatively infects  97% of the total naive population within the first 50 timesteps. However, when the agents are incentivized to self-organize into mobility cohorts, they reduce overall infections by  60%. The rewards averaged across the entire population of agents is shown in Figure 1b. We vary the population's immunity $\delta$ and pandemic fatigue $\alpha$ to see how they affect the average agent's action (Figure 1c). We notice that the *Susceptible* population opts to *'go to less risky location'* or *'stay*

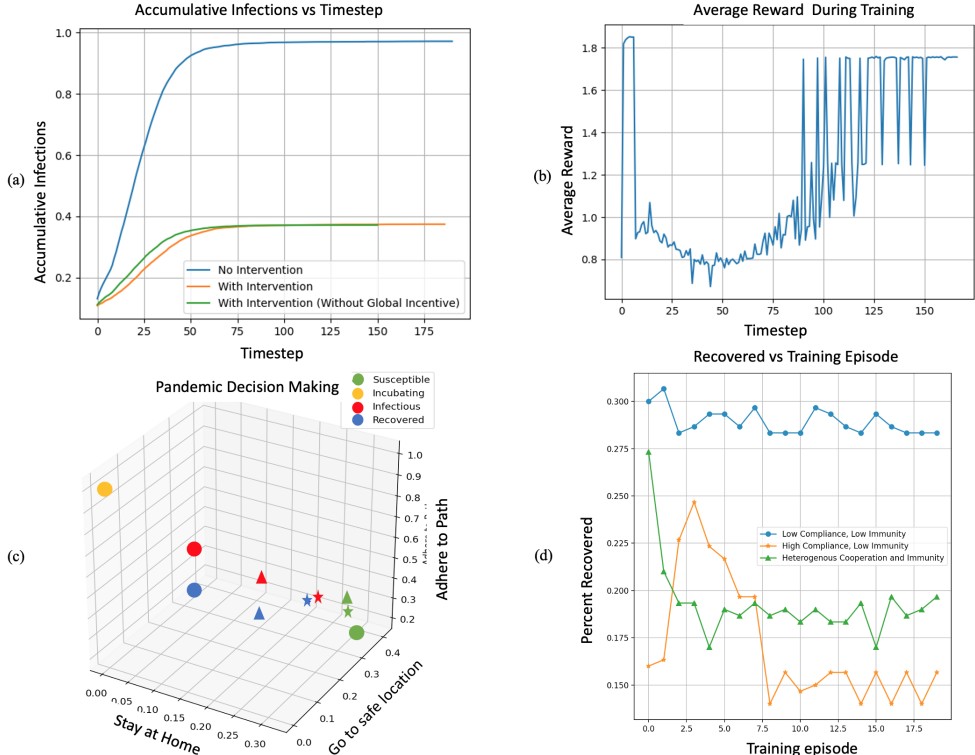

Figure 1: (a) Comparison of mitigation strategies where 'no intervention' refers to agents always going to their next location, 'intervention' refers to agents incentivized by the reward functions in Table 1, and 'intervention without global incentive' refers to agents incentivized by reward functions in Table 1 without $R_{global}$. (b) Average reward variation during training. (c) Average action taken at the end of training. (d) Percentage of *Recovered* agents at the end of a simulation run while varying the skew of heterogenous pandemic fatigue $\alpha$ and immunity $\delta$.

*at home'* the most compared to the other health subgroups (Figure 1c). In contrast, the incubating population, on average, decides to *'go to next location'* 95% of the time which makes sense considering that they cannot infect, nor be infected by others. When the agents become *Infectious*, their collective behavior is most dependent on the population's immunity, as well as their respective pandemic fatigue (Figure 1c). Then, when *Recovered*, they opt to adhere to their own desired destinations less than expected which we believe is because the off-policy algorithm does not have enough timesteps to tweak the policy after acquiring the penalties from being infectious. Furthermore, we observe that the population trained with high compliance (low pandemic fatigue $\delta$) reduces the most infections in comparison to heterogeneous and low compliance (Figure 1d). These initial results are the first step in exploring the feasibility of recommendation systems that mitigate spreading.

## 5 Discussion

Given the the urgency to find non-pharmaceutical interventions, offline multi-agent RL can serve as a safe testbed to experiment with mitigation strategies while benefiting from mobility datasets. We envision a smart phone recommendation system that advises people on how to optimize their mobility during a disease outbreak. Rather than incentivizing people directly, the app's virtual agent learns how to minimize the exposure with respect to the user's social cooperation while collaborating with the other virtual agents to mitigate the disease. We believe this work addresses a largescale RL problem that can benefit both the offline RL and multi-agent RL research communities when fully realized. Our work is currently limited by the fact that we lack ground truth health labels that would otherwise be self reported by app users, therefore we have to rely on disease spreading simulations. Furthermore, our problem exemplifies the challenge of using policy gradient learning on agents whose reward functions change throughout an episode; we leave this for future work.

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
