# OpenReview forum: "On- and Offline Multi-agent Reinforcement Learning for Disease Mitigation using Human Mobility Data "
_NeurIPS.cc/2022/Workshop/Offline_RL — Offline RL Workshop NeurIPS 2022_

### Official Review · Reviewer_yNyf · 2022-10-17
**Missing the mark for me**

**Rating:** 4
**Confidence:** 4

**Review:**

Inspired by Covid-19, authors simulate multiple agents with known disease states and incentives to discourage risky behaviour.  Simulations indicate reductions in predicted disease rates relative to a naive population.

Overall, when I review a paper, I look for either strong theoretical or applied results.  This paper falls into the applied bucket.  For applied papers I look for importance of application domain (this paper looks good in that respect) and practicality of the technique.  On the latter score this paper is tepid at best.  It is unclear how one would obtain the information necessary to deduce disease state (would users ever submit to such surveillance?) or how the incentives would be structured and represented to end users (i.e., reward with lottery tickets?  discount coupons?  threat of incarceration?).  Merely the promise of avoiding a city-wide lockdown is game-theoretically insufficient, everyone will defect unless there is granular feedback.

The simulation results are promising but not particularly surprising in hindsight.  They motivate serious consideration of the concerns in the above paragraph.

---

### Official Review · Reviewer_NEAx · 2022-10-19
**The problem is interesting but lack of technical details**

**Rating:** 5
**Confidence:** 3

**Review:**

Summary:

This paper uses offline multi-agent RL to simulate the spreading of disease infection. The authors used Foursquare location tracking data and made it an RL problem by defining the action space, reward function, and state space with an epidemic model.

Strength:
The problem setting is compelling. Especially, the author has well-defined the action space, reward function, and state space to make the location tracking data to be applicable to an RL problem. The analysis of how immunity and pandemic fatigue will influence the pandemic agent’s behavior is interesting. The work shows a toy example of how offline RL can be used to analyze mobility data and problems.

Weakness:
The main weakness is that the experimental details are insufficient. For example, it is hard to understand what Figure1a is exactly showing. What does the “naive population” stand for (is it a random policy)? And what does the term “intervention” in the figure stand for? In addition, the author mentioned that they used REINFORCE to train the agent but I am a bit concerned about that. Since REINFORCE is generally an on-policy algorithm, is it appropriate to use it in this problem setting? Or did the authors add some specific techniques to make it work? Moreover, it might be nice to include the details of the dataset which will make the whole experiment easier to understand.